# Vaccination Coverage in Hematopoietic Stem Cell Transplant Patients

**DOI:** 10.3390/vaccines13030257

**Published:** 2025-02-28

**Authors:** Angeles Bouzas-Rodríguez, Germán Molina-Romera, Juan Manuel Vázquez-Lago, Olalla Vázquez-Cancela, Cristina Fernández-Pérez

**Affiliations:** 1Department of Preventive Medicine and Public Health, University of Santiago de Compostela, 15786 Santiago de Compostela, Spain; angeles.bouzas.rodriguez@sergas.es (A.B.-R.); juan.manuel.vazquez.lago@sergas.es (J.M.V.-L.); cristina.fernandez.perez3@sergas.es (C.F.-P.); 2Department of Preventive Medicine, Santiago de Compostela University Teaching Hospital, 15706 Santiago de Compostela, Spain; 3Service of Healthy Lifestyles and Health Education, General Subdirectorate of Healthy Lifestyles, 15703 Santiago de Compostela, Spain; german.molina.romera@sergas.es; 4Health Research Institute of Santiago de Compostela, 15706 Santiago de Compostela, Spain

**Keywords:** vaccination coverage, hematopoietic stem cell transplantation, risk groups

## Abstract

**Background/Objectives**: Patients undergoing hematopoietic stem cell transplantation (HSCT) experience profound immunosuppression, increasing their risk of infections. Revaccination is essential to reduce morbidity and mortality. This study aimed to evaluate post-transplant vaccination coverage among patients treated at a specialized reference center. **Methods**: We conducted a cross-sectional, retrospective study including patients who underwent HSCT between 1 January 2018 and 31 May 2021. Vaccination coverage was assessed for each recommended vaccine, and full compliance was defined according to the Spanish Ministry of Health guidelines. A competing risk survival analysis was performed to account for loss to follow-up due to death. Data analysis was carried out using STATA v15. **Results**: Among 138 included patients, 22.46% (31/138) died, and 11.59% (16/138) relapsed. Of the 107 patients who remained in follow-up at 19 months, 41.12% (44/107) (95% CI: 32.26–50.59) had completed the full vaccination schedule, while only 1.87% (2/107) (95% CI: 0.51–6.56) achieved temporal compliance. No significant association was observed between sex and vaccination status or competing risks (*p* > 0.05). **Conclusions**: Post-HSCT vaccination coverage remains suboptimal, highlighting the need for improved vaccination programs, multidisciplinary patient support, and enhanced public and professional awareness to ensure timely immunization in this high-risk population.

## 1. Introduction

Patients receiving immunosuppressive therapies face a significantly increased risk of infectious diseases due to impaired immune function. The expanding indications for these therapies have led to a steady rise in their use across various medical conditions, including autoimmune disorders, organ transplantation, and hematologic malignancies [1]. Among these, hematopoietic stem cell transplantation (HSCT) is a cornerstone treatment for a wide range of congenital and acquired hematological diseases, offering curative potential where conventional therapies fail [2].

Despite substantial advances in HSCT, infectious complications remain a leading cause of morbidity and mortality, often exacerbated by the progressive loss of protective immunity against vaccine-preventable diseases following transplantation [3,4]. Post-HSCT immunodeficiency stems from multiple factors, including the conditioning regimen, the type of transplant (autologous vs. allogeneic), and post-transplant immunosuppressive therapy, which collectively impair immune reconstitution [5,6]. Given these risks, revaccination is a critical yet often overlooked component of post-HSCT care, serving as a fundamental strategy to restore immunity and reduce the burden of severe infections [5,6,7].

Despite international guidelines emphasizing the necessity of revaccination in HSCT recipients, real-world data on vaccination coverage remain scarce, particularly among immunocompromised populations [3,8,9]. Studies from high-income countries suggest that vaccination rates in HSCT recipients are suboptimal, potentially leaving these patients vulnerable to preventable infections [10]. This gap highlights a major disconnect between existing recommendations and clinical practice, underscoring the need for comprehensive assessments of post-HSCT vaccination adherence.

In Spain, the Working Group on Vaccination Programs and Registries endorsed a national strategy in 2018 for the immunization of high-risk patients, including HSCT recipients. However, the extent to which these recommendations are implemented remains unclear, and real-world adherence data are lacking [11]. Without accurate knowledge of vaccination rates in this vulnerable population, it is difficult to design targeted interventions to improve protection against vaccine-preventable diseases.

To address this gap, this study aims to assess and characterize vaccination coverage in a cohort of HSCT recipients at a specialized reference center. By identifying existing disparities, this research seeks to provide a foundation for optimizing post-transplant vaccination strategies and ultimately improving patient outcomes.

## 2. Materials and Methods

### 2.1. Population

Patients who underwent hematopoietic stem cell transplantation (HSCT) at a reference center in the Northwest of Spain for this procedure were included in this study. These reference centers perform HSCT for all patients within the healthcare area who require it, as well as for those from neighboring areas. Therefore, the study population encompasses over 455,000 inhabitants. Inclusion criteria were (i) being over 18 years old and (ii) having undergone HSCT between 1 January 2018 and 31 May 2021 at a reference center.

### 2.2. Data Source

All patients who underwent HSCT were selected through the Business Intelligence tool of the Complex Information and Analysis System (SiAC) during the study period, based on the coding of the hospitalization episode according to the 10th edition of the *International Classification of Diseases* (ICD-10) at the reference center. Vaccination data were obtained from each patient’s electronic medical record (EMR). In the autonomous community of Galicia, there is a vaccination registration system that records all vaccines administered in both public and private centers. This system ensures that all administered vaccines are recorded in a common registry for the entire community. Information regarding the HSCT, its date, clinical course, relapse, or death was obtained from the EMR. Vaccines administered up until 31 December 2022 were considered.

### 2.3. Study Design

This is a cross-sectional study with a retrospective analysis of each patient’s vaccination record. The vaccination schedule of the patients included in this study was compared to the schedule proposed by the Ministry of Health, “Vaccination in risk groups of all ages and in certain situations”, published in 2018 [1].

### 2.4. Outcome Variables

The outcome variables were (i) complete vaccination schedule, regardless of the time of completion, and (ii) complete vaccination schedule adjusted to the recommended timeframe. A vaccination schedule was considered complete when the patient had received at least 3 doses of the 13-valent pneumococcal conjugate vaccine, 1 dose of the 23-valent pneumococcal polysaccharide vaccine, 4 doses of hexavalent vaccine (DTPa/Hib/IPV/HepB) or each of its component vaccines separately, 2 doses of meningococcal ACWY vaccine, and 2 doses of meningococcal B vaccine.

For the time-adjusted analysis, a follow-up period of 19 months was considered. Infections caused by pneumococcus and *Haemophilus influenzae* type b (Hib) are associated with increased severity in these patients, which underscores the importance of revaccination as soon as possible [2].

### 2.5. Vaccination Schedule

According to the Ministry of Health recommendations, the vaccination schedule should be completed starting 18 months after the date of HSCT. Given the complexity of post-transplant care and the potential for clinical complications, an additional month was added to this target date to allow for possible delays in scheduling and any medical conditions that may temporarily contraindicate vaccination, such as ongoing immunosuppressive therapy or active infections.

Additionally, the data were individually analyzed for each vaccine included in the complete vaccination schedule (Appendix A). To ensure a structured follow-up and reflect real-world vaccination practices, three key time points were established: 6, 13, and 19 months after HSCT. These intervals were selected based on the recommended vaccination timeline, allowing an assessment of coverage at different stages of the vaccination process.

Since vaccination should begin three months after HSCT, any vaccine doses administered before this period were excluded from the analysis and considered as not administered within the appropriate timeframe.

#### Exclusion Criteria for Certain Vaccines

COVID-19 and influenza vaccines were excluded from the assessment of the complete vaccination schedule. These vaccines are administered through seasonal mass vaccination campaigns and are recommended for the general population, not exclusively for HSCT recipients. Additionally, coverage for these vaccines is generally higher due to widespread public health initiatives, making them less informative when evaluating vaccination adherence specific to post-HSCT patients.

Other vaccines, such as live attenuated vaccines (e.g., MMR, varicella, yellow fever), were excluded because their administration depends on the patient’s immune status and is contraindicated in cases of prolonged immunosuppression. The HPV (human papillomavirus) and hepatitis A vaccines were also excluded, as they are only indicated in specific situations, rather than being universally recommended for all HSCT recipients.

Thus, the final analysis of vaccination coverage was conducted using vaccines recommended for all HSCT patients, ensuring comparability across the study population (Appendix A).

### 2.6. Statistical Analysis

A competing risks survival analysis was performed to assess the probability of completing the full vaccination schedule and receiving individual vaccines within the recommended timeframe. Competing risks arise when patients are censored due to events that preclude further follow-up, such as
(i)Death before the completion of the vaccination schedule.(ii)Relapse requiring a second HSCT, which resets the vaccination timeline.(iii)Relocation to another autonomous community, leading to loss of access to vaccination records.

#### 2.6.1. Survival Model Specification

A Cox proportional hazards model, utilizing the likelihood ratio test, was used to estimate the cumulative incidence function (CIF) for vaccination coverage among vaccinated versus unvaccinated individuals. This approach provides a more accurate assessment than traditional Kaplan–Meier methods, which may overestimate the probability of event occurrence in the presence of competing risks.

The model included the following covariates:

Patient age at transplantation, categorized into clinically relevant age groups to account for potential differences in immune response and adherence to follow-up.

Sex, given that previous studies suggest potential sex-related differences in healthcare-seeking behavior and vaccination adherence.

#### 2.6.2. Software and Statistical Significance

All statistical analyses were conducted using STATA (version 15), and a *p*-value < 0.05 was considered statistically significant. Results were reported with 95% confidence intervals (CIs) to reflect the precision of the estimates.

### 2.7. Ethical Regulation

This study was approved by the Territorial Committee of Ethics in Research of Santiago-Lugo (Registration number 2022/259). This study was conducted in accordance with Good Clinical Practice guidelines and the ethical principles outlined in the Declaration of Helsinki (Fortaleza 2013). Data handling complied with the Spanish Organic Law 3/2018 on Personal Data Protection and Digital Rights, ensuring strict confidentiality measures. All patient data were pseudonymized, maintaining a clear technical and functional separation between the research team and the entity responsible for pseudonymization (the Health Area’s Subdirectorate of Information Systems). This process guarantees that researchers do not have access to directly identifiable patient information.

Given the retrospective nature of this study and the use of pseudonymized data, obtaining individual informed consent was not required. In accordance with national regulations, specifically Additional Provision 17 of Organic Law 3/2018, the use of clinical records for research purposes is permitted without explicit patient consent, provided that the study is carried out by healthcare professionals and follows institutional approval procedures, ensuring compliance with data protection regulations. This legal framework supports the use of pseudonymized data for research while maintaining patient confidentiality and adherence to ethical and legal standards.

## 3. Results

A total of 138 patients who underwent HSCT between 1 January 2018 and 31 May 2021 were included in this study. Of these, 48.6% (67/138) were women, with a mean age of 53.32 years (SD 13.31). During the study period, 22.5% (31/138) of patients died, and 11.6% (16/138) experienced a relapse, requiring a second HSCT. In total, 151 HSCT procedures were analyzed. No patients were lost to follow-up due to transfers outside the autonomous community’s healthcare system.

To monitor vaccination adherence, we selected follow-up time points at 6, 13, and 19 months post-HSCT. These intervals align with the typical vaccination schedule recommendations for immunocompromised patients and the expected timeframes for immune system recovery after HSCT. Additionally, these periods allow for the assessment of early and late mortality, as well as potential relapses that may influence vaccination completion. Table 1 presents the number of patients remaining in follow-up at each time point.

Of the 107 patients still under follow-up at 19 months, 8.41% (9/107) had not received any vaccine dose, while 50.47% (54/107) initiated the vaccination schedule but did not complete it. The median time from HSCT discharge to the first vaccine was 182.9 days (IQR: 111–288). Among patients who initiated but did not complete the vaccination schedule, PCV was the most frequently administered vaccine, followed by PPSV23.

### 3.1. Complete Vaccination Schedule

At 19 months post-HSCT, 41.1% (44/107) (95% CI: 32.3–50.6) of patients had completed the full vaccination schedule according to the Ministry of Health’s recommendations. However, when considering the recommended timeline for at-risk groups, only 1.87% (2/107) (95% CI: 0.5–6.6) met the criterion within the expected period. The low adherence suggests potential barriers such as patient-related factors (e.g., treatment side effects, healthcare access) or system-level challenges (e.g., scheduling delays, provider awareness).

### 3.2. Pneumococcal Vaccination

Completion of the full pneumococcal vaccination schedule (3 PCV + PPSV23) was observed in 56.6% (64/113) (95% CI: 47.4–65.5) of patients with at least 13 months of follow-up. However, only 5.3% (6/113) (95% CI: 2.5–11.10) received it within the recommended timeframe.

For the pneumococcal conjugate vaccine (PCV), 71.4% (85/119) (95% CI: 62.7–78.8) of patients received at least one dose during the study period. However, only 12.6% (15/119) (95% CI: 7.8–19.8) received it within the first 6 months post-HSCT.

Regarding the pneumococcal polysaccharide vaccine (PPSV23), coverage reached 67.3% (76/113) (95% CI: 58.2–75.2), but only 9.7% (11/113) (95% CI: 5.52–16.59) received it within the first 13 months.

### 3.3. Hexavalent Vaccination

Hexavalent vaccine coverage was 47.7% (51/107) (95% CI: 38.5–57.0). When considering the recommended completion timeline (within 19 months post-HSCT), only 16.82% (18/107) (95% CI: 10.9–25) met this criterion.

### 3.4. Meningococcal ACWY Vaccination

Coverage for meningococcal ACWY vaccination was 66.36% (71/107) (95% CI: 57–74.6), with only 43% (46/107) (95% CI: 34.0–52.5) of HSCT patients receiving the vaccination within the first 19 months post-transplant.

### 3.5. Meningococcal B Vaccination

Meningococcal B vaccine coverage was 61.7% (66/107) (95% CI: 52.2–70.3). Based on the recommended timeline, coverage was 31.8% (34/107) (95% CI: 23.7–41.1) of HSCT patients who completed the 19-month follow-up.

### 3.6. Survival Analysis

A competing risks regression model was used to assess the relationship between age and sex with time to complete vaccination. The Wald chi-square for the model was 3.3 (*p* = 0.3501), indicating a non-significant association (Figure 1).

No significant differences were found between sex and the risk of complete vaccination or competing events (death or relapse) (*p* > 0.05). However, patients aged 45–65 showed a trend toward a lower likelihood of completing vaccination compared to those under 45, though this association did not reach statistical significance (*p* = 0.119).

The lack of statistically significant findings suggests that factors beyond age and sex—such as socioeconomic status, healthcare access, and post-HSCT complications—may play a larger role in vaccination adherence. Future studies should explore these variables to identify potential interventions.

## 4. Discussion

This study is the first to specifically assess vaccination coverage in high-risk patients who have undergone hematopoietic stem cell transplantation (HSCT). Our findings reveal significant gaps in vaccination coverage: 8.41% of patients who completed at least 19 months of follow-up had not received any vaccines, while 50.5% initiated vaccination but did not complete the schedule. Only 41.1% of patients received the full vaccination series. Notably, when adjusting coverage to align with the optimal guidelines recommended by the Spanish Ministry of Health, only 1.9% of patients met all recommended non-live vaccine requirements within the 19-month post-transplant period.

These results underscore a critical issue: patients at high risk for severe infections remain unprotected for extended periods, which may contribute to increased morbidity, mortality, and healthcare utilization. Previous studies have reported similarly low vaccination rates among HSCT recipients, highlighting that suboptimal immunization coverage is a persistent challenge in this population [3]. Unlike vaccination programs targeting the general population, which set clear benchmarks (e.g., 75% for influenza and COVID-19, 90% for diphtheria in infancy [4]), there are no universally established targets for immunization in high-risk groups. However, given that the goal in immunocompromised patients is individual rather than herd immunity, coverage should ideally exceed the targets set for the general population. The fact that less than 2% of our study cohort met national vaccination recommendations suggests that many HSCT recipients remain inadequately protected for longer than recommended, with potential clinical consequences [3].

Live attenuated vaccines (e.g., MMR, varicella) were excluded from this study because their administration depends on the patient’s immune status and is contraindicated in cases of prolonged immunosuppression. Data on the safety of MMR and varicella vaccines in hematopoietic transplant recipients within the first two years post-transplantation are limited. Current guidelines recommend their use only in seronegative individuals, at least two years post-transplant, and under strict conditions: absence of immunosuppressive treatment for one year, no systemic immunoglobulins for 8–11 months, and no graft-versus-host disease (GVHD) [5,6]. During this period, the immune system remains vulnerable, and live-attenuated vaccines pose a significant risk of uncontrolled viral replication and severe disease. Their interaction with immunosuppressive treatments further increases the likelihood of adverse effects. For these reasons, live-attenuated vaccines were excluded from this study [7,8].

Conversely, for inactivated vaccines, no reasons have been found to delay vaccination, except in situations of temporary or absolute contraindications due to vaccine incompatibility. In this regard, there is some controversy regarding how to proceed with patients receiving immunosuppressive treatment. It may be advisable to wait between 3 to 6 months after the end of treatment to improve vaccine effectiveness. However, even in these cases, vaccination might still be recommended, considering the risk–benefit ratio for the patient. For instance, the CDC suggests that live attenuated vaccines should not be administered for at least 3 months after immunosuppressive therapy, while non-live vaccines administered during chemotherapy should be readministered after immune competence is regained [9].

Overall, while live attenuated vaccines are contraindicated during periods of significant immunosuppression due to safety concerns, inactivated vaccines can often be administered, with timing adjusted based on individual patient circumstances to optimize efficacy and safety. The fact that at-risk patients do not benefit from vaccination may lead to an increase in infectious processes, relapse, and higher mortality rates [1,9,10].

### 4.1. Potential Barriers to Vaccination

Several factors may contribute to the observed low vaccination rates. One key barrier is the lack of systematic referral to vaccination services. HSCT patients often receive care from multiple specialists, and if vaccination is not explicitly prioritized by their primary provider, opportunities for immunization may be missed. Previous studies have highlighted that vaccination uptake improves when clinicians actively discuss and facilitate immunization during routine follow-up care [11,12]. The low referral rate to preventive medicine services in our cohort suggests a need for better integration of vaccination within post-transplant care protocols.

Another critical factor is insufficient patient education and awareness. While our study does not directly assess patient attitudes toward vaccination, previous research indicates that concerns about vaccine safety, particularly in immunocompromised individuals, can contribute to hesitancy [13,14]. Additionally, some patients may not recall receiving clear recommendations from their physicians, further reducing uptake. Addressing these gaps requires a structured approach, including educational interventions tailored to both patients and healthcare providers.

The complexity of vaccination schedules also plays a role. In our study, the hexavalent vaccine had the lowest completion rate, likely due to the requirement for multiple doses. Patients undergoing HSCT frequently require hospital visits for monitoring and treatment, yet adherence to multi-dose vaccines remains challenging. Simplified schedules, such as single-dose formulations where available, could improve compliance. For instance, the recent introduction of the 20-valent pneumococcal vaccine may enhance coverage by reducing the number of required doses [15]. In our study, the most frequently administered vaccine was the conjugate pneumococcal vaccine. This vaccine is recommended for all types of at-risk patients, from those with chronic conditions or over 65 years of age to those with a high level of immunosuppression, such as patients undergoing HSCT.

Financial and logistical barriers may also influence vaccine uptake. Our findings indicate that adherence to MenACWY vaccination was higher than to MenB vaccination, which may be related to differences in funding and accessibility. In our community, the MenB vaccine is only funded for specific high-risk situations and requires individualized approval from public health authorities, whereas the MenACWY vaccine is more readily available in hospital and preventive medicine services [16]. This highlights the need for policy-level interventions to ensure equitable access to all recommended vaccines for HSCT patients.

### 4.2. Clinical Implications

In our study, 8.4% of patients did not initiate vaccination. The consequences of suboptimal vaccination coverage in HSCT recipients are significant, as these patients face a substantially elevated risk of severe infections, including an 80% increased risk of invasive pneumococcal disease and higher rates of hospitalization and mortality compared to healthy individuals [16]. Delays in immunization prolong susceptibility to vaccine-preventable diseases, increasing the likelihood of avoidable complications and contributing to higher healthcare costs.

Our findings highlight the urgency of implementing targeted strategies to improve vaccination rates in this vulnerable population. Integrating automatic reminders and standardized vaccination pathways into electronic health records could help ensure timely immunization, reducing missed opportunities for vaccination. Additionally, enhanced physician training is essential to emphasize the importance of vaccination and improve communication with patients regarding its benefits and safety. Improving accessibility to vaccines, particularly those requiring multiple doses, through expanded funding and streamlined approval processes, would further facilitate adherence. Finally, patient education campaigns aimed at addressing concerns about vaccine safety and efficacy—especially among immunocompromised individuals—are crucial to overcoming hesitancy and increasing uptake. Strengthening these approaches could significantly enhance vaccination coverage and, consequently, improve patient outcomes.

### 4.3. Limitations

Our study has several limitations. First, we did not assess the specific reasons why some patients did not initiate or complete vaccination. While previous research suggests that lack of physician recommendation and vaccine hesitancy may be contributing factors, further qualitative studies are needed to explore these barriers in detail. Second, our analysis focused on vaccination uptake rather than clinical outcomes. Moreover, we were unable to provide detailed clinical information, such as the types of HSCT, graft-versus-host disease (GVHD), and concomitant treatments, which could influence vaccination adherence and outcomes. The absence of these data limits the interpretation of vaccination adherence in this context, and this should be considered when evaluating the findings. Due to practical reasons and the scarcity of published data, current guidelines and protocols recommend the same vaccinations regardless of the type of transplant (autologous or allogenic), the source of hematopoietic progenitors (peripheral blood, bone marrow, or umbilical cord), and the conditioning regimen received [5]. Future research should investigate whether incomplete immunization is associated with higher infection rates in HSCT recipients. Third, our study was conducted in a single region, which may limit the generalizability of our findings to other healthcare settings with different vaccination policies. Additionally, the representativeness of the study population may be limited, as the sample size and the characteristics of the patients included may not fully reflect the broader population of HSCT recipients. However, it is important to note that this study was carried out in a tertiary care center that treats HSCT patients from other areas, which may partially mitigate this limitation. Furthermore, all patients who underwent HSCT in our center during the study years were included, ensuring a comprehensive evaluation within our setting. Finally, in the survival analysis, we only considered age and sex as variables, without adjusting for other potential confounders that could influence vaccination adherence and its impact on patient outcomes. Future studies should incorporate a broader range of clinical and sociodemographic factors to provide a more comprehensive understanding of the determinants of vaccination coverage in this high-risk population.

## 5. Conclusions

This study highlights persistently low vaccination coverage among HSCT recipients, with significant gaps in adherence to national immunization recommendations. Addressing this issue requires a multifaceted approach, including better integration of vaccination into post-transplant care, improved patient and physician education, and policy-level changes to enhance vaccine accessibility. Given the high risk of severe infections in this population, optimizing vaccination strategies should be a priority to improve patient outcomes and reduce preventable morbidity and mortality.

## Figures and Tables

**Figure 1 vaccines-13-00257-f001:**
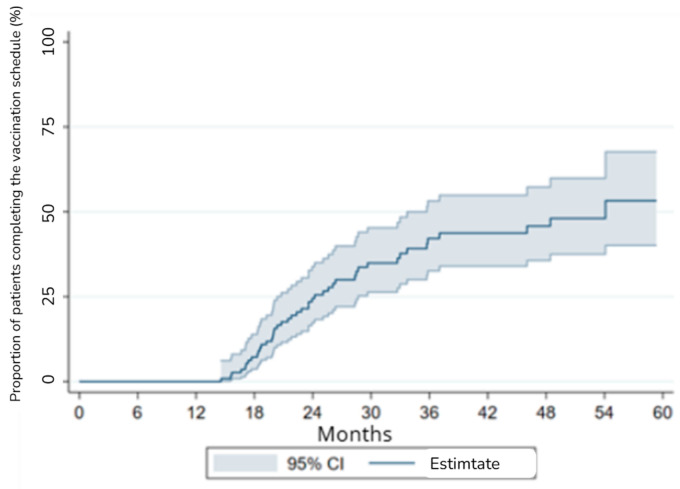
Survival analysis with competing risks for vaccination coverage.

**Table 1 vaccines-13-00257-t001:** Patients continuing in this study at follow-up months.

	Processes	Exitus	Relapse
Start	151	0	0
6 months	119	23	9
13 months	113	4	2
19 months	107	4	2

## Data Availability

Data are available upon request.

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
