# Peer review of "Vaccination Coverage in Hematopoietic Stem Cell Transplant Patients"

_vaccines, 2025, doi:10.3390/vaccines13030257_

Round 1

Reviewer 1 Report

Comments and Suggestions for Authors

The abstract is structurally incomprehensible, methodologically vague, and superficially statistical. The abstract should have clearly indicated its scope, results, or any other meaningful outcome of the observations. Serious grammatical errors and loss of valuable data points logically make it not credible as a scientific summary.

The introduction is lacking in focus, depth, and an exciting scientific rationale. It also gives extremely superficial background on knowledge gap framing or defining the importance of the study. The accuracy of the specific terms, flow of the existing literature, and the abrupt transitions further weaken it. Without the more detailed and structured justification given, the current study appears descriptive and does not address a clinically meaningful question.

Materials and Methods are not rigorous, transparent, or coherent, suffering from imprecise definitions, arbitrary methodological choices, and poor justification for some key decisions. There is no obvious reason for segmenting the follow-up periods into 7, 13, and 19 months. The exclusion of COVID-19 and influenza vaccines is not well supported.

All the exclusion criteria shall be specified.
Competing risks survival analysis is poorly described.
A limitation is that only age and sex are included in the model.
Handling of loss to follow-up is not well described.

Ethical considerations are superficial-there is a lack of discussion on informed consent and the assurance of privacy for the patients. Unless extensively revised, the methodology is neither reproducible nor scientifically sound.

The Results section is statistically weak, and lacks critical interpretation. The arbitrary choice of time points for follow-up and incomplete description of vaccination uptake assess its impact difficult to make. Also, the underpowered survival analysis does not offer any meaningful insights. Major revisions are required for clarity, coherence, and scientific rigor in this section. Table 1 and the figure 1 are not fully integrated into the manuscript.

The Discussion is scientifically weak, lacking in depth, and lacking engagement with the critical issues; the overused novelty, indefinite comparisons, mechanisms that are far from explanations, and lack of contextualization-the scientific contribution stands highly limited in this study. Rather than just restating findings, the Discussion should have critiqued what went wrong-so low vaccination rates-applying perspectives from both a structural and a clinical point. No comparison is given from previous work, as this text is thick with general information rather than digging deep into in-depth analysis to review the current literature. Second, the analysis of barriers is not detailed while speculations toward physician education go superficially, and there's no strong reasoning for prioritization of vaccines as well, to make the given study relevant in application. The final limitation section is poorly developed and does not consider the real methodological limitations, further reducing the strength of the study.

The conclusion sounds more like superficial statements that would apply to almost any vaccination study in an immunocompromised population rather than a forward-looking, evidence-based discussion.

Comments on the Quality of English Language

Careful language revision by a native English-speaking person or by a professional editor is highly recommended.

Author Response

Comment1: The abstract is structurally incomprehensible, methodologically vague, and superficially statistical. The abstract should have clearly indicated its scope, results, or any other meaningful outcome of the observations. Serious grammatical errors and loss of valuable data points logically make it not credible as a scientific summary.

Response1: We regret that the abstract did not clearly convey the key aspects of our study. Based on your feedback, we have revised its structure to enhance readability, clarified the methodological details, and ensured that the scope and main findings are more explicitly presented. Additionally, we have carefully reviewed the language to address any grammatical issues and improve overall coherence. We appreciate your constructive comments, as they have helped us refine the quality and clarity of our manuscript.

Now, the text may read as follows:

Background/Objectives: Patients undergoing hematopoietic stem cell transplantation (HSCT) experience profound immunosuppression, increasing their risk of infections. Revaccination is essential to reduce morbidity and mortality. This study aimed to evaluate post-transplant vaccination coverage among patients treated at a specialized reference center.  Methods: We conducted a cross-sectional, retrospective study including patients who underwent HSCT between January 1, 2018, and May 31, 2021. Vaccination coverage was assessed for each recommended vaccine and full compliance was defined according to the Spanish Ministry of Health guidelines. A competing-risk survival analysis was performed to account for loss to follow-up due to death. Data analysis was carried out using STATA v15. Results: Among 138 included patients, 22.46% (31/138) died, and 11.59% (16/138) relapsed. Of the 107 patients who remained in follow-up at 19 months, 41.12% (44/107) (95% CI: 32.26–50.59) had completed the full vaccination schedule, while only 1.87% (2/107) (95% CI: 0.51–6.56) achieved temporal compliance. No significant association was observed between sex and vaccination status or competing risks (p > 0.05).  Conclusions: Post-HSCT vaccination coverage remains suboptimal, highlighting the need for improved vaccination programs, multidisciplinary patient support, and enhanced public and professional awareness to ensure timely immunization in this high-risk population.

Comment2: The introduction is lacking in focus, depth, and an exciting scientific rationale. It also gives extremely superficial background on knowledge gap framing or defining the importance of the study. The accuracy of the specific terms, flow of the existing literature, and the abrupt transitions further weaken it. Without the more detailed and structured justification given, the current study appears descriptive and does not address a clinically meaningful question.

Response 2:Thank you for your insightful feedback on the introduction. We acknowledge your concerns regarding focus, depth, and scientific rationale, as well as the need for a more structured justification and smoother transitions.

In response, we have thoroughly revised the introduction to enhance clarity, strengthen the rationale, and provide a more detailed discussion of the knowledge gap. We have explicitly highlighted the importance of revaccination in HSCT recipients, emphasizing the high burden of infectious complications and the suboptimal vaccination coverage observed in immunocompromised patients. Additionally, we have improved the flow of the literature review, ensuring a logical progression of ideas and eliminating abrupt transitions.

Now, the text may read as follows:

Patients receiving immunosuppressive therapies face a significantly increased risk of infectious diseases due to impaired immune function. The expanding indications for these therapies have led to a steady rise in their use across various medical conditions, including autoimmune disorders, organ transplantation, and hematologic malignancies [1]. Among these, hematopoietic stem cell transplantation (HSCT) is a cornerstone treatment for a wide range of congenital and acquired hematological diseases, offering curative potential where conventional therapies fail [2].

Despite substantial advances in HSCT, infectious complications remain a leading cause of morbidity and mortality, often exacerbated by the progressive loss of protective immunity against vaccine-preventable diseases following transplantation [3,4]. Post-HSCT immunodeficiency stems from multiple factors, including the conditioning regimen, the type of transplant (autologous vs. allogeneic), and post-transplant immunosuppressive therapy, which collectively impair immune reconstitution [5]. Given these risks, revaccination is a critical yet often overlooked component of post-HSCT care, serving as a fundamental strategy to restore immunity and reduce the burden of severe infections [6].

Despite international guidelines emphasizing the necessity of revaccination in HSCT recipients, real-world data on vaccination coverage remain scarce, particularly among immunocompromised populations [7,8]. Studies from high-income countries suggest that vaccination rates in HSCT recipients are suboptimal, potentially leaving these patients vulnerable to preventable infections [9,10]. This gap highlights a major disconnect between existing recommendations and clinical practice, underscoring the need for comprehensive assessments of post-HSCT vaccination adherence.

In Spain, the Working Group on Vaccination Programs and Registries endorsed a national strategy in 2018 for the immunization of high-risk patients, including HSCT recipients. However, the extent to which these recommendations are implemented remains unclear, and real-world adherence data are lacking [11]. Without accurate knowledge of vaccination rates in this vulnerable population, it is difficult to design targeted interventions to improve protection against vaccine-preventable diseases.

To address this gap, this study aims to assess and characterize vaccination coverage in a cohort of HSCT recipients at a specialized reference center. By identifying existing disparities, this research seeks to provide a foundation for optimizing post-transplant vaccination strategies and ultimately improving patient outcomes.

We believe these modifications now provide a clearer and more compelling justification for the study, aligning it with a clinically meaningful research question. We appreciate your valuable input and hope that the revised version meets the expectations of the journal.

Comment 3 : Materials and Methods are not rigorous, transparent, or coherent, suffering from imprecise definitions, arbitrary methodological choices, and poor justification for some key decisions. There is no obvious reason for segmenting the follow-up periods into 7, 13, and 19 months. The exclusion of COVID-19 and influenza vaccines is not well supported.

All the exclusion criteria shall be specified.
Competing risks survival analysis is poorly described.
A limitation is that only age and sex are included in the model.
Handling of loss to follow-up is not well described.

Response 3: Thank you for your valuable feedback on the Materials and Methods section. We understand your concerns regarding the lack of rigor, transparency, and coherence in the methodological details. We agree that the section can be improved for better clarity, and we will revise it to ensure the methodology is more transparent, coherent, and logically structured.

Regarding the specific concerns you raised:

Follow-up periods (7, 13, and 19 months):

The choice of these time points was based on the Spanish Ministry of Health protocol that we used as a guide for classifying patients as vaccinated or not vaccinated. This protocol recommends the 6, 12, and 18-month marks as the ideal time for administering vaccines post-transplant. However, these recommendations only account for the "ideal" administration time and do not consider potential challenges in clinical practice, such as scheduling issues or patient-related factors (e.g., fever, patient absences, etc.) that could prevent timely vaccination. To mitigate these potential issues, we allowed a margin of one month to ensure patients could still be included in the vaccination coverage calculation despite delays. This approach ensures a more realistic and representative assessment of vaccination coverage in everyday clinical practice. If we were to strictly adhere to 6, 12, and 18 months, we believe the vaccination coverage rate reported in the manuscript would be significantly lower, which would not accurately reflect the real-world situation.

We will revise the Methods section to explicitly reference the Ministry of Health protocol and clarify the rationale for this segmentation.

Exclusion of COVID-19 and influenza vaccines:

The decision not to include COVID-19 and influenza vaccines in our study was based on the fact that in our autonomous community (region), COVID-19 and influenza vaccination campaigns are managed through primary care and are targeted to different population groups, such as the general public or other specific high-risk groups. These vaccines are primarily administered at large vaccination centers and not in the hospital setting, which further distinguishes them from the vaccines specifically related to transplant patients. Consequently, the vaccination coverage for these vaccines is presumed to be higher than for the vaccines specific to hematopoietic stem cell transplantation (HSCT), which are managed through specialized healthcare services and focus on a more limited and defined high-risk population (i.e., transplant patients).

Additionally, including these vaccines could introduce bias when comparing vaccination coverage, as they follow a different distribution system with potentially greater accessibility. We will clarify these points in the revised manuscript.

Specification of exclusion criteria:

We acknowledge the need for greater transparency in listing all exclusion criteria. We will ensure that all exclusion criteria are explicitly specified in the revised Methods section, including those related to specific vaccines and any other relevant considerations that might impact the analysis.

Competing risks survival analysis:

We recognize that our description of the competing risks survival analysis requires more detail. We will expand this section to clarify the statistical approach used, including how competing risks (death, relapse requiring a new HSCT, and loss to follow-up) were accounted for in the model.

Limitation of only including age and sex in the model:

We acknowledge this limitation and will explicitly mention it in the discussion section. While additional factors such as comorbidities or socioeconomic status could be relevant, these data were not systematically available in our dataset.

Handling of loss to follow-up:

Loss to follow-up was addressed using a competing risks survival model, where patients who died, relapsed requiring a new HSCT, or moved to another region were censored at the time of their last available data point. We will clarify this methodology in the revised Methods section.

To ensure these points are understood, following your recommendations, we have modified the Materials and Methods section. It now reads as follows:

2.5 Vaccination Schedule

According to the Ministry of Health recommendations, the vaccination schedule should be completed starting 18 months after the date of HSCT. Given the complexity of post-transplant care and the potential for clinical complications, an additional month was added to this target date to allow for possible delays in scheduling and any medical conditions that may temporarily contraindicate vaccination, such as ongoing immunosuppressive therapy or active infections.

Additionally, the data were analyzed individually for each vaccine included in the complete vaccination schedule (Appendix A). To ensure a structured follow-up and reflect real-world vaccination practices, three key time points were established: 6, 13, and 19 months after HSCT. These intervals were selected based on the recommended vaccination timeline, allowing an assessment of coverage at different stages of the vaccination process.

Since vaccination should begin three months after HSCT, any vaccine doses administered before this period were excluded from the analysis and considered as not administered within the appropriate timeframe.

Exclusion Criteria for Certain Vaccines

COVID-19 and influenza vaccines were excluded from the assessment of the complete vaccination schedule. These vaccines are administered through seasonal mass vaccination campaigns and are recommended for the general population, not exclusively for HSCT recipients. Additionally, coverage for these vaccines is generally higher due to widespread public health initiatives, making them less informative when evaluating vaccination adherence specific to post-HSCT patients.

Other vaccines, such as live attenuated vaccines (e.g., MMR, varicella, yellow fever), were excluded because their administration depends on the patient's immune status and is contraindicated in cases of prolonged immunosuppression. The HPV (human papillomavirus) and hepatitis A vaccines were also excluded, as they are only indicated in specific situations, rather than being universally recommended for all HSCT recipients.

Thus, the final analysis of vaccination coverage was conducted using vaccines recommended for all HSCT patients, ensuring comparability across the study population (Appendix A).

2.6 Statistical Analysis

A competing risks survival analysis was performed to assess the probability of completing the full vaccination schedule and receiving individual vaccines within the recommended timeframe. Competing risks arise when patients are censored due to events that preclude further follow-up, such as:

  1. i) Death before the completion of the vaccination schedule.
  2. ii) Relapse requiring a second HSCT, which resets the vaccination timeline.

iii) Relocation to another Autonomous Community, leading to loss of access to vaccination records.

Survival Model Specification

A competing risks model was used to estimate the cumulative incidence function (CIF) for vaccination coverage, providing a more accurate assessment than traditional Kaplan-Meier methods, which may overestimate the probability of event occurrence in the presence of competing risks.

The model included the following covariates:

Patient age at transplantation, categorized into clinically relevant age groups to account for potential differences in immune response and adherence to follow-up.

Sex, given that previous studies suggest potential sex-related differences in healthcare-seeking behavior and vaccination adherence.

Software and Statistical Significance

All statistical analyses were conducted using STATA (version 15), and a p-value < 0.05 was considered statistically significant. Results were reported with 95% confidence intervals (CIs) to reflect the precision of the estimates.

We appreciate your thoughtful comments and believe these revisions will significantly improve the clarity and rigor of our manuscript.

Comment 4:Ethical considerations are superficial-there is a lack of discussion on informed consent and the assurance of privacy for the patients. Unless extensively revised, the methodology is neither reproducible nor scientifically sound.

Response 4:Thank you for your feedback regarding the ethical considerations of our study. We acknowledge the need to provide a more detailed discussion on informed consent and data privacy to ensure clarity and transparency.

Our study complies with Good Clinical Practice guidelines and follows the ethical principles established in the latest update of the Declaration of Helsinki (Fortaleza 2013). Regarding data protection, the study adheres to the Spanish Organic Law 3/2018 on Personal Data Protection and Digital Rights, which mandates strict safeguards to ensure patient confidentiality.

Specifically:

Data Pseudonymization and Privacy Assurance:

All data are pseudonymized following the legal requirements, ensuring a strict separation between the research team and the entity responsible for pseudonymization. The Health Area’s Subdirectorate of Information Systems is responsible for pseudonymizing the data and maintaining the reidentification keys, ensuring that researchers do not have access to directly identifiable patient information. This approach complies with the legal framework and protects patient privacy.

Informed Consent:

Given the retrospective nature of this study and the use of pseudonymized data, individual informed consent was not obtained. However, the study design complies with national regulations, which allow the use of anonymized or pseudonymized data for research purposes without explicit patient consent, provided that robust data protection measures are in place.

We have revised the Ethical Considerations section of the manuscript to explicitly include these details, ensuring a more comprehensive discussion of patient privacy and informed consent. It now reads as follows:

The study was approved by the Territorial Committee of Ethics in Research of Santiago-Lugo (Registration number 2022/259). This study was conducted in accordance with Good Clinical Practice guidelines and the ethical principles outlined in the Declaration of Helsinki (Fortaleza 2013). Data handling complied with the Spanish Organic Law 3/2018 on Personal Data Protection and Digital Rights, ensuring strict confidentiality measures. All patient data were pseudonymized, maintaining a clear technical and functional separation between the research team and the entity responsible for pseudonymization (the Health Area’s Subdirectorate of Information Systems). This process guarantees that researchers do not have access to directly identifiable patient information.

Given the retrospective nature of the study and the use of pseudonymized data, obtaining individual informed consent was not required. In accordance with national regulations, specifically Additional Provision 17 of Organic Law 3/2018, the use of clinical records for research purposes is permitted without explicit patient consent, provided that the study is carried out by healthcare professionals and follows institutional approval procedures, ensuring compliance with data protection regulations. This legal framework supports the use of pseudonymized data for research while maintaining patient confidentiality and adherence to ethical and legal standards.

We appreciate your comments, which help us improve the clarity and rigor of our ethical considerations.

Comment 5:The Results section is statistically weak, and lacks critical interpretation. The arbitrary choice of time points for follow-up and incomplete description of vaccination uptake assess its impact difficult to make. Also, the underpowered survival analysis does not offer any meaningful insights. Major revisions are required for clarity, coherence, and scientific rigor in this section. Table 1 and the figure 1 are not fully integrated into the manuscript.

Response 5: We regret that you had that impression, and we truly appreciate your insightful feedback. Following your suggestions, we have carefully revised the Results section to improve its clarity, consistency, and scientific rigor. It now reads as follows:

At 19 months post-HSCT, 41.12% (44/107) (95% CI: 32.26–50.59) of patients had completed the full vaccination schedule according to the Ministry of Health's recommendations. However, when considering the recommended timeline for at-risk groups, only 1.87% (2/107) (95% CI: 0.51–6.56) met the criterion within the expected period. The low adherence suggests potential barriers such as patient-related factors (e.g., treatment side effects, healthcare access) or system-level challenges (e.g., scheduling delays, provider awareness)

3.2 Pneumococcal Vaccination

Completion of the full pneumococcal vaccination schedule (3 PCV + PPSV23) was observed in 56.64% (64/113) (95% CI: 47.43–65.51) of patients with at least 13 months of follow-up. However, only 5.31% (6/113) (95% CI: 2.46–11.10) received it within the recommended timeframe.

For the pneumococcal conjugate vaccine (PCV), 71.43% (85/119) (95% CI: 62.74–78.78) of patients received at least one dose during the study period. However, only 12.61% (15/119) (95% CI: 7.79–19.76) received it within the first 6 months post-HSCT.

Regarding the pneumococcal polysaccharide vaccine (PPSV23), coverage reached 67.26% (76/113) (95% CI: 58.16–75.22), but only 9.73% (11/113) (95% CI: 5.52–16.59) received it within the first 13 months.

3.3 Hexavalent Vaccination

Hexavalent vaccine coverage was 47.66% (51/107) (95% CI: 38.45–57.04). When considering the recommended completion timeline (within 19 months post-HSCT), only 16.82% (18/107) (95% CI: 10.91–25.03) met this criterion.

3.4 Meningococcal ACWY Vaccination

Coverage for meningococcal ACWY vaccination was 66.36% (71/107) (95% CI: 56.97 - 74.60), with only 42.99% (46/107) (95% CI: 34.01 - 52.45) of HSCT patients receiving the vaccination within the first 19 months post-transplant.

3.5 Meningococcal B Vaccination

Meningococcal B vaccine coverage was 61.68% (66/107) (95% CI: 52.22 - 70.34). Based on the recommended timeline, coverage was 31.78% (34/107) (95% CI: 23.72 - 41.10) of HSCT patients who completed the 19-month follow-up.

3.6 Survival Analysis

A competing risks regression model was used to assess the relationship between age and sex with time to complete vaccination. The Wald chi-square for the model was 3.28 (p = 0.3501), indicating a non-significant association (Figure 1).

No significant differences were found between sex and the risk of complete vaccination or competing events (death or relapse) (p > 0.05). However, patients aged 45–65 showed a trend toward a lower likelihood of completing vaccination compared to those under 45, though this association did not reach statistical significance (p = 0.119).

The lack of statistically significant findings suggests that factors beyond age and sex—such as socioeconomic status, healthcare access, and post-HSCT complications—may play a larger role in vaccination adherence. Future studies should explore these variables to identify potential interventions.

Your comments have been invaluable in strengthening our study and ensuring that our findings are presented more robustly. We are grateful for your time and effort in reviewing our work, and we sincerely appreciate your constructive input.

Comment 6:The Discussion is scientifically weak, lacking in depth, and lacking engagement with the critical issues; the overused novelty, indefinite comparisons, mechanisms that are far from explanations, and lack of contextualization-the scientific contribution stands highly limited in this study. Rather than just restating findings, the Discussion should have critiqued what went wrong-so low vaccination rates-applying perspectives from both a structural and a clinical point. No comparison is given from previous work, as this text is thick with general information rather than digging deep into in-depth analysis to review the current literature. Second, the analysis of barriers is not detailed while speculations toward physician education go superficially, and there's no strong reasoning for prioritization of vaccines as well, to make the given study relevant in application. The final limitation section is poorly developed and does not consider the real methodological limitations, further reducing the strength of the study.

Response 6: We sincerely appreciate your detailed feedback on the Discussion section. We understand your concerns regarding the need for a more in-depth analysis of the low vaccination rates and a stronger contextualization from both structural and clinical perspectives.

To address these issues, we have revised the Discussion in several key areas:

  1. Explaining low vaccination rates: We have expanded our analysis of the factors contributing to suboptimal vaccination coverage, incorporating both structural barriers (e.g., lack of standardized vaccination pathways, limited accessibility to certain vaccines, and absence of reminder systems) and clinical barriers (e.g., insufficient prioritization by healthcare providers and potential uncertainties regarding vaccine administration in immunocompromised patients).
  2. Comparison with previous studies: We have included additional references to better contextualize our findings within the existing literature on vaccination in hematopoietic stem cell transplant recipients. This allows for a clearer understanding of similarities and differences with prior research.
  3. More detailed analysis of barriers: We have elaborated on the potential reasons behind incomplete vaccination schedules, distinguishing between healthcare system-related factors and physician- and patient-related perceptions.
  4. Justification for vaccine prioritization: We have clarified why certain vaccines have higher coverage rates than others, discussing how factors such as the number of required doses, public funding, and vaccine availability may influence adherence.
  5. Strengthening the limitations section: We have provided a more detailed discussion of the methodological limitations of our study, particularly regarding the variables included in the survival analysis (which only considered age and sex). Additionally, we have highlighted the need for further research to assess the clinical impact of vaccination in this population.

It now reads as follows:

This study is the first to specifically assess vaccination coverage in high-risk patients who have undergone hematopoietic stem cell transplantation (HSCT). Our findings reveal significant gaps in vaccination coverage: 8.41% of patients who completed at least 19 months of follow-up had not received any vaccines, while 50.47% initiated vaccination but did not complete the schedule. Only 41.12% of patients received the full vaccination series. Notably, when adjusting coverage to align with the optimal guidelines recommended by the Spanish Ministry of Health, only 1.87% of patients met all recommended non-live vaccine requirements within the 19-month post-transplant period.

These results underscore a critical issue: patients at high risk for severe infections remain unprotected for extended periods, which may contribute to increased morbidity, mortality, and healthcare utilization. Previous studies have reported similarly low vaccination rates among HSCT recipients, highlighting that suboptimal immunization coverage is a persistent challenge in this population[12]. Unlike vaccination programs targeting the general population, which set clear benchmarks (e.g., 75% for influenza and COVID-19, or 90% for diphtheria in infancy [13]) there are no universally established targets for immunization in high-risk groups. However, given that the goal in immunocompromised patients is individual rather than herd immunity, coverage should ideally exceed the targets set for the general population. The fact that less than 2% of our study cohort met national vaccination recommendations suggests that many HSCT recipients remain inadequately protected for longer than recommended, with potential clinical consequences [12].

On the contrary, no reasons have been found to delay vaccination, except in situations of temporary or absolute contraindications due to vaccine incompatibility. In this regard, there is some controversy regarding how to act with patients receiving immunosuppressive treatment, where it may be advisable to wait between 3 to 6 months after the end of treatment to improve vaccine effectiveness. However, even in these cases, vaccination might still be recommended, considering the risk-benefit ratio for the patient [14]. The fact that at-risk patients do not benefit from vaccination may lead to an increase in infectious processes, relapses, and higher mortality rates [11,14,15].

Potential Barriers to Vaccination

Several factors may contribute to the observed low vaccination rates. One key barrier is the lack of systematic referral to vaccination services. HSCT patients often receive care from multiple specialists, and if vaccination is not explicitly prioritized by their primary provider, opportunities for immunization may be missed. Previous studies have highlighted that vaccination uptake improves when clinicians actively discuss and facilitate immunization during routine follow-ups care [19-20]. The low referral rate to preventive medicine services in our cohort suggests a need for better integration of vaccination within post-transplant care protocols.

Another critical factor is insufficient patient education and awareness. While our study does not directly assess patient attitudes toward vaccination, previous research indicates that concerns about vaccine safety, particularly in immunocompromised individuals, can contribute to hesitancy [17,18]. Additionally, some patients may not recall receiving clear recommendations from their physicians, further reducing uptake. Addressing these gaps requires a structured approach, including educational interventions tailored to both patients and healthcare providers.

The complexity of vaccination schedules also plays a role. In our study, the hexavalent vaccine had the lowest completion rate, likely due to the requirement for multiple doses. Patients undergoing HSCT frequently require hospital visits for monitoring and treatment, yet adherence to multi-dose vaccines remains challenging. Simplified schedules, such as single-dose formulations where available, could improve compliance. For instance, the recent introduction of the 20-valent pneumococcal vaccine may enhance coverage by reducing the number of required doses [21]. In our study, the most frequently administered vaccine was the conjugate pneumococcal vaccine. This vaccine is recommended for all types of at-risk patients, from those with chronic conditions or over 65 years of age to those with a high level of immunosuppression, such as patients undergoing HSCT.

Financial and logistical barriers may also influence vaccine uptake. Our findings indicate that adherence to MenACWY vaccination was higher than to MenB vaccination, which may be related to differences in funding and accessibility. In our Community, the MenB vaccine is only funded for specific high-risk situations and requires individualized approval from public health authorities, whereas the MenACWY vaccine is more readily available in hospital and preventive medicine services [22].This highlights the need for policy-level interventions to ensure equitable access to all recommended vaccines for HSCT patients.

Clinical Implications

In our study, 8.41% of patients did not initiate vaccination. The consequences of suboptimal vaccination coverage in HSCT recipients are significant, as these patients face a substantially elevated risk of severe infections, including an 80% increased risk of invasive pneumococcal disease and higher rates of hospitalization and mortality compared to healthy individuals [16]. Delays in immunization prolong susceptibility to vaccine-preventable diseases, increasing the likelihood of avoidable complications and contributing to higher healthcare costs.

Our findings highlight the urgency of implementing targeted strategies to improve vaccination rates in this vulnerable population. Integrating automatic reminders and standardized vaccination pathways into electronic health records could help ensure timely immunization, reducing missed opportunities for vaccination. Additionally, enhanced physician training is essential to emphasize the importance of vaccination and improve communication with patients regarding its benefits and safety. Improving accessibility to vaccines, particularly those requiring multiple doses, through expanded funding and streamlined approval processes, would further facilitate adherence. Finally, patient education campaigns aimed at addressing concerns about vaccine safety and efficacy—especially among immunocompromised individuals—are crucial to overcoming hesitancy and increasing uptake. Strengthening these approaches could significantly enhance vaccination coverage and, consequently, improve patient outcomes.

Limitations

Our study has several limitations. First, we did not assess the specific reasons why some patients did not initiate or complete vaccination. While previous research suggests that lack of physician recommendation and vaccine hesitancy may be contributing factors, further qualitative studies are needed to explore these barriers in detail. Second, our analysis focused on vaccination uptake rather than clinical outcomes; future research should investigate whether incomplete immunization is associated with higher infection rates in HSCT recipients. Third, our study was conducted in a single region, which may limit the generalizability of our findings to other healthcare settings with different vaccination policies. Finally, in the survival analysis, we only considered age and sex as variables, without adjusting for other potential confounders that could influence vaccination adherence and its impact on patient outcomes. Future studies should incorporate a broader range of clinical and sociodemographic factors to provide a more comprehensive understanding of the determinants of vaccination coverage in this high-risk population.

We hope these revisions strengthen the Discussion and enhance the scientific contribution of our study. We truly appreciate your insightful comments and remain open to any further suggestions that could improve the clarity and rigor of our manuscript.

Comment 7:The conclusion sounds more like superficial statements that would apply to almost any vaccination study in an immunocompromised population rather than a forward-looking, evidence-based discussion.

Response 7:Thank you for your valuable feedback; we have revised the conclusion to focus more specifically on our study's findings and to provide a forward-looking, evidence-based discussion tailored to the unique challenges faced by HSCT recipients.

Following your recommendations, we have modified the Materials and Methods section. It now reads as follows:

This study highlights persistently low vaccination coverage among HSCT recipients, with significant gaps in adherence to national immunization recommendations. Addressing this issue requires a multifaceted approach, including better integration of vaccination into post-transplant care, improved patient and physician education, and policy-level changes to enhance vaccine accessibility. Given the high risk of severe infections in this population, optimizing vaccination strategies should be a priority to improve patient outcomes and reduce preventable morbidity and mortality.

We are very grateful for your thorough review and insightful recommendations. Your suggestions have greatly contributed to enhancing the quality and rigor of our manuscript. We have revised the text accordingly, and we hope that the updated version reflects these improvements and meets your expectations.

Reviewer 2 Report

Comments and Suggestions for Authors

It is well known that patients undergoing hematopoietic stem cell transplantation (HSCT) are at risk of infectious complications, which determines the expediency of preventing these complications, including through the use of vaccines. The aim of the study was to describe the vaccination coverage in a group of patients who underwent HSCT in one of the centers of this procedure. At the same time, the aim of the study was limited in content because the authors used insufficiently representative samples of patients to determine the effect of vaccination on quality of life, the presence of infectious complications and survival of patients who underwent HSCT. In addition, the following disadvantages and limitations of this study can be identified:

(1) The authors considered only standard vaccines used for the prevention of pneumococcal infections, as well as the use of a hexavalent vaccine (DTPa/Hib/IPV/HepB - for the prevention of diphtheria, tetanus, pertussis, polio, hemophilus influenzae type b and hepatitis B), the use of which covered patients of all groups. However, it should be noted that the authors mentioned this limitation in Section 4. Discussion.

(2) The studied groups of patients (with and without vaccination, who died during the data collection period) were not clinically characterized, including: the presence of infectious complications and graft-versus-host reactions, as well as types of HSCT (allogeneic or auto-HSCT), conditioning regimens used (myeloablative and non-myeloablative), the presence of intensive concomitant therapy (antibiotics, antiviral, antifungal drugs, parenteral nutrition). It should be noted that vaccinations are only one component of a complex of anti-infective therapy and prevention in patients who have undergone HSCT. Apparently, the authors do not have this information, which is important for the characterization of patient groups.

(3) Section 2.6 Statistical Analysis. In this section, authors should report methods for detecting differences between groups (chi-squared, etc.) and a confidence level (p<0.05), whether or not there are significant differences in the study results.

(4) Figure 1. It is advisable to indicate what the ordinate scale means.

(5) Subsection Limits (lines 282-287). It should be noted that the low representativeness of the studied patient groups did not allow to confirm or deny the usefulness of using certain types of vaccines for survival and the presence of infectious complications in patients after HSCT. This circumstance limits the practical significance of the presented research.

Author Response

Comment1:It is well known that patients undergoing hematopoietic stem cell transplantation (HSCT) are at risk of infectious complications, which determines the expediency of preventing these complications, including through the use of vaccines. The aim of the study was to describe the vaccination coverage in a group of patients who underwent HSCT in one of the centers of this procedure. At the same time, the aim of the study was limited in content because the authors used insufficiently representative samples of patients to determine the effect of vaccination on quality of life, the presence of infectious complications and survival of patients who underwent HSCT. In addition, the following disadvantages and limitations of this study can be identified:

  • The authors considered only standard vaccines used for the prevention of pneumococcal infections, as well as the use of a hexavalent vaccine (DTPa/Hib/IPV/HepB - for the prevention of diphtheria, tetanus, pertussis, polio, hemophilus influenzae type b and hepatitis B), the use of which covered patients of all groups. However, it should be noted that the authors mentioned this limitation in Section 4. Discussion.

Response 1:Thank you for your comment. Indeed, we focused on the standard vaccines used for preventing pneumococcal infections and the hexavalent vaccine due to the specific context of our study. The decision not to include COVID-19 and influenza vaccines was based on the fact that these vaccines are managed through primary care in our region, targeting broader population groups, and are administered primarily in large vaccination centers. In contrast, vaccines specific to hematopoietic stem cell transplantation (HSCT) are managed within specialized healthcare services for a more defined high-risk population. This distinction led us to focus on the vaccines directly related to transplant patients. We appreciate your understanding of this decision, which is mentioned in the Discussion section.

Comment 2:The studied groups of patients (with and without vaccination, who died during the data collection period) were not clinically characterized, including: the presence of infectious complications and graft-versus-host reactions, as well as types of HSCT (allogeneic or auto-HSCT), conditioning regimens used (myeloablative and non-myeloablative), the presence of intensive concomitant therapy (antibiotics, antiviral, antifungal drugs, parenteral nutrition). It should be noted that vaccinations are only one component of a complex of anti-infective therapy and prevention in patients who have undergone HSCT. Apparently, the authors do not have this information, which is important for the characterization of patient groups.

We sincerely appreciate your insightful comment. Unfortunately, the data regarding infectious complications, graft-versus-host reactions, types of HSCT, and intensive concomitant therapies were not available in the database we managed for this study. While we fully recognize the importance of these factors in a comprehensive analysis, we believe that the main findings of our study regarding low vaccination coverage remain significant and relevant, regardless of the clinical characteristics. We are grateful for your suggestion and hope to explore these aspects further in future research if additional data becomes available. Thank you again for your valuable input

Comment 3: Section 2.6 Statistical Analysis. In this section, authors should report methods for detecting differences between groups (chi-squared, etc.) and a confidence level (p<0.05), whether or not there are significant differences in the study results.

Response 3:Thank you for your helpful suggestion. We have expanded Section 2.6. We hope this addition provides a clearer understanding of the statistical approach employed in the study. It now reads as follows:

A competing risks survival analysis was performed to assess the probability of completing the full vaccination schedule and receiving individual vaccines within the recommended timeframe. Competing risks arise when patients are censored due to events that preclude further follow-up, such as:

  1. i) Death before the completion of the vaccination schedule. ii) Relapse requiring a second HSCT, which resets the vaccination timeline. iii) Relocation to another Autonomous Community, leading to loss of access to vaccination records.

Survival Model Specification

A competing risks model was used to estimate the cumulative incidence function (CIF) for vaccination coverage, providing a more accurate assessment than traditional Kaplan-Meier methods, which may overestimate the probability of event occurrence in the presence of competing risks.

The model included the following covariates:

Patient age at transplantation, categorized into clinically relevant age groups to account for potential differences in immune response and adherence to follow-up.

Sex, given that previous studies suggest potential sex-related differences in healthcare-seeking behavior and vaccination adherence.

Software and Statistical Significance

All statistical analyses were conducted using STATA (version 15), and a p-value < 0.05 was considered statistically significant. Results were reported with 95% confidence intervals (CIs) to reflect the precision of the estimates.

Comment 4: Figure 1. It is advisable to indicate what the ordinate scale means.

Response 4: You are absolutely right to ask for clarification. The vertical axis represents the proportion of patients who complete the vaccination. We will make these definitions clearer in the figure legend.

Comment 5: Subsection Limits (lines 282-287). It should be noted that the low representativeness of the studied patient groups did not allow to confirm or deny the usefulness of using certain types of vaccines for survival and the presence of infectious complications in patients after HSCT. This circumstance limits the practical significance of the presented research.

Response 5: Thank you for your valuable feedback. You are absolutely right, and we have added this consideration to the limitations section of our manuscript to emphasize the impact of the low representativeness of the studied patient groups on the ability to confirm or deny the usefulness of certain vaccines for survival and infectious complications after HSCT.

Now, the text may read as follows:

Our study has several limitations. First, we did not assess the specific reasons why some patients did not initiate or complete vaccination. While previous research suggests that lack of physician recommendation and vaccine hesitancy may be contributing factors, further qualitative studies are needed to explore these barriers in detail. Second, our analysis focused on vaccination uptake rather than clinical outcomes; future research should investigate whether incomplete immunization is associated with higher infection rates in HSCT recipients. Third, our study was conducted in a single region, which may limit the generalizability of our findings to other healthcare settings with different vaccination policies. Finally, in the survival analysis, we only considered age and sex as variables, without adjusting for other potential confounders that could influence vaccination adherence and its impact on patient outcomes. Future studies should incorporate a broader range of clinical and sociodemographic factors to provide a more comprehensive understanding of the determinants of vaccination coverage in this high-risk population.

Thank you for your thoughtful review and constructive feedback. Your comments have been extremely helpful in improving the manuscript, and we have carefully addressed each of your suggestions. We appreciate your time and expertise, and we hope that the revisions make the study clearer and more impactful.

Reviewer 3 Report

Comments and Suggestions for Authors

This is an important study

2.5 Vaccination schedule

According to the recommendations from the Ministry of Health, the vaccination schedule should be completed starting 18 months after the date of HSCT. An additional month was added to this target date to allow for possible delays in scheduling and any
clinical conditions that may prevent vaccination

[Readers will be interested to know the evidence based studies that support the recommendation for 18 months]

Other vaccines, such as live attenuated vaccines (which depend on the clinical status of the patient) and the HPV (human papillomavirus) and hepatitis A vaccines, which are only indicated in certain situations, were also excluded to allow for individualized cases.

[Readers will be interested to know why live attenuated vaccines were excluded as they could have been analysed separately and are of interest]

143 “8.55% (67/138” 

[Your sample is not large enough to use decimals to 2 places]

3.1 Completed vaccination schedule

“Of the patients who remained in the study after 19 months, 41.12% (44/107) (95% CI: 32.26 - 50.59) completed the vaccination schedule according to the Ministry of Health's 60
recommendations”

“This is the first study on vaccination coverage in high-risk groups that focuses specifically on patients who have undergone HSCT. We observed that 8.41% of patients who completed at least 19 months of follow-up had not received any vaccines. Additionally, 50.47% initiated vaccination schedules but did not complete them. Only 41.12% of patients received the complete vaccination schedule. The most frequently administered
vaccine was the pneumococcal conjugate vaccine

“One of the reasons for limited access to vaccination is the low uptake of patients due to
the lack of referrals from the clinical services responsible for patients undergoing HSCT
to vaccination units, as well as the low priority given to vaccination in patient care [19].”

[this is a low vaccination rate for such a key population. Please recommend methods from your literature review you could use to encourage vaccination and not depend on low referral rates? Interdepartment research teams? or working groups? Readers will be interested].

======

Figure 1 What does Stimation mean?

Figure 1 please define the vertical axis

Typo

1.59% (16/138) .. Of the 107 patients who remained in the study at 19 months, 41.12% (44/107) (95% CI 32.26–50.59) completed vaccination, and 1.87% (2/107) (95% CI 0.51–6.56) achieved temporal compliance. relapsed.  [word relapsed is misplaced]

Author Response

Comment 1:2.5 Vaccination schedule

“According to the recommendations from the Ministry of Health, the vaccination schedule should be completed starting 18 months after the date of HSCT. An additional month was added to this target date to allow for possible delays in scheduling and any
clinical conditions that may prevent vaccination”

[Readers will be interested to know the evidence based studies that support the recommendation for 18 months]

Response 1:Thank you very much for your valuable comment. This recommendation is based on the clinical management guidelines from the Spanish Ministry of Health, which we have referenced in the text as suggested. These guidelines indicate the minimum time that should pass between the first vaccination and HSCT, and they emphasize that vaccines should be administered as soon as possible within this timeframe. When patients receive all the recommended vaccines while respecting the minimum intervals between doses, the inactivated vaccination schedule is completed by 18 months, which is the reason for selecting this target date. We greatly appreciate your feedback and hope this clarifies the rationale behind our approach

Comment 2: “Other vaccines, such as live attenuated vaccines (which depend on the clinical status of the patient) and the HPV (human papillomavirus) and hepatitis A vaccines, which are only indicated in certain situations, were also excluded to allow for individualized cases.”

[Readers will be interested to know why live attenuated vaccines were excluded as they could have been analysed separately and are of interest]

Response 2: Thank you for your insightful comment. The aim of our study is to assess whether HSCT patients are receiving appropriate vaccination coverage, regardless of their clinical evolution or individual characteristics. For this reason, we focused only on the vaccination coverage for the vaccines common to all HSCT patients, without considering variables such as age, clinical progression, sex, or underlying conditions.

While live attenuated vaccines and others like the HPV and hepatitis A vaccines are indeed of interest, their exclusion was intentional to maintain consistency in the analysis and ensure that all patients, regardless of their individual circumstances, were included in the evaluation. For example, the HPV vaccine is only funded in our region for women under 45 years old, the hepatitis A vaccine is indicated only for individuals with liver pathology, and live attenuated vaccines are administered based on clinical evolution (treatment) and serological results. We hope this clarification addresses your concern, and we appreciate your feedback.

Comments 3:143 “8.55% (67/138” 

[Your sample is not large enough to use decimals to 2 places]

Response 3:Thank you for your valuable feedback. We have made the adjustment as suggested and revised the figure to reflect whole numbers rather than decimals. We appreciate your attention to detail.

Comments 4: 3.1 Completed vaccination schedule

“Of the patients who remained in the study after 19 months, 41.12% (44/107) (95% CI: 32.26 - 50.59) completed the vaccination schedule according to the Ministry of Health's 60
recommendations”

“This is the first study on vaccination coverage in high-risk groups that focuses specifically on patients who have undergone HSCT. We observed that 8.41% of patients who completed at least 19 months of follow-up had not received any vaccines. Additionally, 50.47% initiated vaccination schedules but did not complete them. Only 41.12% of patients received the complete vaccination schedule. The most frequently administered
vaccine was the pneumococcal conjugate vaccine

“One of the reasons for limited access to vaccination is the low uptake of patients due to
the lack of referrals from the clinical services responsible for patients undergoing HSCT
to vaccination units, as well as the low priority given to vaccination in patient care [19].”

[this is a low vaccination rate for such a key population. Please recommend methods from your literature review you could use to encourage vaccination and not depend on low referral rates? Interdepartment research teams? or working groups? Readers will be interested].

Response 4:Thank you for your valuable feedback. We have revised the discussion to include specific strategies for improving vaccination uptake in HSCT patients. Based on our literature review and your suggestion, we now propose the implementation of interdisciplinary research teams and working groups to foster better coordination between clinical services and vaccination units. Additionally, we recommend targeted education programs for healthcare providers, the integration of automated reminders in electronic health records to ensure timely referrals, the creation of dedicated vaccination pathways for transplant patients, and patient education campaigns to address concerns and raise awareness about the importance of vaccination in immunocompromised populations. We believe these strategies will help improve vaccination rates in this vulnerable group.

Comments 5: Figure 1 What does Stimation mean?

Figure 1 please define the vertical axis

Response 5:You are absolutely right to ask for clarification. The vertical axis represents the proportion of patients who complete the vaccination. The term "Stimation" is a mistake; the correct term should be "Estimation" or, more precisely, "Projection" or "Estimate", as it refers to the predicted proportion of patients expected to complete their full vaccination over the months. We will make these definitions clearer in the figure legend.

Comments 6:1.59% (16/138) .. Of the 107 patients who remained in the study at 19 months, 41.12% (44/107) (95% CI 32.26–50.59) completed vaccination, and 1.87% (2/107) (95% CI 0.51–6.56) achieved temporal compliance. relapsed.  [word relapsed is misplaced]

Response 6:Thank you for pointing that out. You are correct that the word "relapsed" is misplaced in this context. Here is the revised version:

Comments 7:" Of the 107 patients still under follow-up at 19 months, 8.41% (9/107) had not received any vaccine dose, while 50.47% (54/107) initiated the vaccination schedule but did not complete it.

Response 7:We have removed the word "relapsed" to ensure the sentence is clear and accurate.

We sincerely appreciate the time and effort you have dedicated to reviewing our manuscript. Your insightful comments and suggestions have been instrumental in improving the quality and clarity of our work. We have carefully addressed each point, and we hope that the revised version meets your expectations and enhances the overall contribution of the study.
